# Comparison of the Optical Properties of Different Dielectric Materials (SnO₂, ZnO, AZO, or SiAlNₓ) Used in Silver-Based Low-Emissivity Coatings

**Ana Cueva * and Enrique Carretero**

Departamento de Física Aplicada, Universidad de Zaragoza, C/Pedro Cerbuna 12, 50009 Zaragoza, Spain; ecarre@unizar.es
* Correspondence: ana.cueva@unizar.es

**Abstract:** This work analyzed and compared the optical and photoenergetic properties of low-emissivity coatings made from various dielectric materials deposited through magnetron sputtering following a systematic, comparable method. Different multilayer structures of silver-based low-emissivity coatings were studied using $SnO_2$, $ZnO$, $SiAlN_x$, and aluminum-doped zinc oxide (AZO, which is inherently a semiconductor, but it fulfils an optical dielectric function in this type of structure). The properties of the coatings were determined by spectrophotometric and sheet resistance measurements. Coatings with AZO as the dielectric layers obtain the best photoenergetic performance because silver growth is more efficient on AZO. We also studied the effect of ion bombardment on AZO and $SiAlN_x$ in an attempt to obtain a better low-emissivity coating, achieving better results when etching the dielectric layer with an ion gun. Regarding the structures' visible transmission, the oxides produced better transmission results. Based on the above, we concluded that AZO had the best optical and photoenergetic properties in our deposition system, observing, in the best-case scenario, improvements in emissivity from 0.083 with $SnO_2$ to 0.058 with AZO and to 0.052 using an ion beam on AZO and improvements in visible transmission from 81.9% with $SnO_2$ to 86.8% with AZO.

**Keywords:** sputtering; thin films; low-emissivity coatings; dielectric materials; optical properties



## 1. Introduction

Low-emissivity (low-e) and solar control coatings are essential to improve the energy efficiency of glazing. These coatings are widely used and play a versatile role in architectural and automotive glass [1–3]. On the one hand, the aim is to produce coatings with high transmittance in the visible spectral range to take full advantage of the visible light passing through the glazing. On the other hand, the goal is to maximize solar reflectance, which refers to the proportion of solar radiation reflected by the glazing, to prevent too much energy from entering the glazed area. Finally, these kinds of coatings have a low emissivity at room temperature, which means that they have a high reflectance in the emission zone of the black body, translating into greater comfort inside the glazed area by reducing both energy losses in winter and solar gain in summer. Therefore, the overall objective is to obtain a coating with high visible transmittance, high solar reflectance, and low emissivity [4,5].

Low-emissivity layers can be obtained with a single layer of aluminum-doped zinc oxide (AZO) or indium tin oxide (ITO) (all the abbreviations in the manuscript have been listed in Table S1 of the Supplementary Material), which are transparent, conductive materials; however, given that greater selectivity is required in the solar range, we did not set out to study these materials in monolayers. Multilayer stacks with a noble metal layer have been used to obtain low-e coatings with the desired characteristics, including high reflectivity in the mid- and far-infrared regions. The noble metal layer is sandwiched between

two dielectric layers that provide antireflective properties in the visible spectrum since the structure can suppress the reflection of the metal in the visible region, thus achieving a high transmission in this range [6–10]. Furthermore, the dielectric layers provide the noble metal layer with chemical and mechanical protection; they act as an adhesive and nucleation layer [11]; and they also serve as a barrier against diffusion processes [12,13]. One of the most important large-area coating technologies is physical vapor deposition (PVD), and the most common PVD technique is sputtering [14–18]. It consists of bombarding a material, called the target, with ions that evaporate the target material and deposit it on the surface of the substrate, generating a metal film or a compound if a reactive gas is used. The layers are deposited sequentially, so if an oxide-based dielectric layer is deposited on top of the noble metal, the latter needs protection from oxidation. A very thin metal barrier layer must therefore be deposited on top of the noble metal layer. The barrier layer, which could be Ti, Ni, or Cr, is just 2–3 nm thick, so it does not significantly alter the structure's overall properties [19,20]. Fundamental aspects of these types of materials deposited in thin films can be explained by theories such as Time-Dependent Density Functional Theory, among others [21–24].

The aforementioned multilayer structures, such as substrate/dielectric/metal/barrier/ dielectric, where the dielectric layer is a metal oxide or nitride deposited by sputtering, have excellent properties in terms of heat insulation, solar energy reflection, visible transparency, and electrical conductivity. As such, they are widely used in products that incorporate low-e coatings in insulating glass units (IGUs) and solar control coatings for automotive windshields [25]. Materials such as Ag, Au, and Cu are used as a highly reflective metal layer [26–30]. The most widely used metal is silver, as gold is more expensive and copper is more reactive and produces unattractive colors in architecture. Moreover, Ag exhibits high reflectance in the mid- and far-infrared regions and high electrical conductivity, and it has lower optical absorption in the visible range [31]. The choice of the dielectric material in the structure is critical to obtain low emissivities. There are a lot of studies in the literature about the properties of multiple tri-layer structures (dielectric/metal/dielectric) with different dielectric materials: $SnO_2$ [32,33], ZnO [34–38], AZO (aluminum-doped zinc oxide) [39–44], ITO (tin-doped indium oxide) [45,46] (AZO and ITO are semiconductor materials but they fulfil an optical dielectric function in this type of structure), ZnS [47,48], TiN [49,50], $TiO_2$ [51], TiAlN (Al-doped titanium nitride) [52], AlN [53], and SiN [19,54]. Furthermore, comparative studies have also been conducted among some of them [55–59]. Each of these materials has its advantages and disadvantages for industrial use. For example, $SnO_2$ is the most widely used material because of its high deposition rate, although it does not have a good affinity with Ag, obtaining higher emissivities than those obtained with other materials, such as ZnO, which provides better emissivities and a good deposition rate [11]. SiN is extensively employed in the glass industry due to its excellent mechanical and thermal protection properties [19,20], but it provides higher emissivity when deposited before Ag. $TiO_2$ is also widely used in glazing applications for its high transmittance in the visible spectrum and good chemical stability [51], but it has worse emissivity and lower deposition rate.

The use of an ion gun to pre-smooth the dielectric layer before silver deposition has also been studied to achieve lower emissivities because silver grows in a more crystalline structure when deposited on dielectric layers that have previously been treated with an ion gun [60].

As explained above, the basic structure of low-emissivity coatings is a tri-layer structure with a metal sandwiched between two dielectric layers. However, some applications require higher selectivity in the visible range, as well as higher insulation and solar control. This is why metal double-layer structures (dielectric/metal/dielectric/metal/dielectric) are widely used by glass manufacturers [1,5].

This study aimed to compare the optical and photoenergetic properties of single-silver structures (dielectric/Ag/Ti/dielectric) and double-silver structures (dielectric/Ag/Ti/ dielectric/Ag/Ti/dielectric) made with different dielectric materials. The coatings were

all deposited using the same sputtering deposition system and under the same conditions (which were adjusted to achieve an optimal deposition rate and a process pressure similar to that used in industrial processes), with the same amount of silver and dielectric material, and using the same substrate to obtain totally comparable emissivity and visible transmittance results. The materials assessed in this study were $SnO_2$, ZnO, AZO, AZO deposited in an argon plus oxygen atmosphere, and $SiAlN_x$. We also used an ion gun to etch the dielectric layer before silver deposition and examine the resulting impact on the optical and photoenergetic properties of the low-emissivity structures [60]. We selected these dielectric materials for their wide use in the architectural glass industry and because: $SnO_2$ has a high deposition rate [5]; ZnO exhibits a stronger affinity for silver than $SnO_2$ during deposition [11]; AZO (Al-doped ZnO) provides a higher barrier against the diffusion of Na atoms contained in the glass substrate [12] and higher moisture resistance [55], and $SiAlN_x$ produces low-e coatings with higher mechanical strengths and thermal resistances [19,20].

All mentioned studies have been conducted by sputtering in different deposition systems, with various parameters, and under different deposition conditions. Our study aimed to homogenize and standardize all these results and be able to compare the emissivity and visible transmission of low-emissivity coatings deposited using different materials in the dielectric layers. In the construction and automotive industries, emissivity is an important parameter with respect to thermal insulation and a high visible transmission improves visibility, hence the focus of this study. We used different quantities of silver in our low-emissivity coatings to cover a range of applications that require high levels of visible transmission, neutral colors in reflection, and sufficient solar energy reflection and thermal insulation to produce thermally comfortable glazing.

## 2. Experiment

### 2.1. Preparation of Low-e Multilayer Films

We used a semi-industrial, inline, high vacuum magnetron sputtering deposition system to produce all the samples in this study (see Figure S1a,b of the Supplementary Material), operating with 600 mm × 100 mm rectangular targets (see Figures S2 and S3 of the Supplementary Material). The system worked at a base pressure of $7 \times 10^{-7}$ mbar in the process chamber before introducing the process gases, and the working pressure was $1$–$2 \times 10^{-3}$ mbar, depending on the material deposited and the number of process gases injected into the system (in this deposition system, a flow of 200 standard cubic centimeters per minute (sccm) of Ar was approximately equivalent to a pressure of $1 \times 10^{-3}$ mbar). The deposition was performed at room temperature. The deposition system used a pulsed DC source and a mass flow controller (MFC) to control the amount of gas injected into the system. This system operated dynamically, which means that the substrate moved at a constant speed throughout the process chamber while the deposition was taking place, producing uniform coatings of homogeneous thicknesses. To obtain substrates free from impurities, we incorporated an ion gun to produce a cleaner substrate surface and improve layer adhesion [61]. The deposition system was also equipped with a cryogenic pump to reduce the residual water vapor. We used low-iron sodium calcium float glass (Pilkington Optiwhite™), substrates measuring 100 mm × 100 mm and 4 mm thick, previously cleaned with a special glass detergent (ACEDET 5509).

Single-metal samples (dielectric/metal/barrier/dielectric) and double-metal samples (dielectric/metal/barrier/dielectric/metal/barrier/dielectric) containing Ag metal layers and Ti barrier layers were deposited using the same high vacuum deposition system. Different dielectric materials were tested to compare their photoenergetic properties: $SnO_2$, ZnO, AZO, AZO deposited under argon and a small amount of $O_2$ (called AZO_2), and $SiAlN_x$. Combinations of AZO and $SiAlN_x$ were also attempted on the same structure. Finally, we used an ion gun to etch the dielectric layer, which was calibrated for the etching thickness of each material. Table 1 presents the deposition conditions of the different materials, where the $O_2$ and $N_2$ flows were calibrated to work in the reactive mode to obtain $SnO_2$, ZnO, and $SiAlN_x$. Tables 2 and 3 present the thicknesses and the different

materials in the layers of the 29 single-Ag samples (S1 to S12) and 10 double-Ag samples (D1 to D5) (see the FE-SEM cross-section of Sample S1B in Figure S4 of the Supplementary Material). The samples were categorized according to the type of dielectric material in the structure. Additionally, they were labeled based on the amount of silver in the multilayer, so we could compare structures with an equivalent silver content (e.g., S1A is an S1 single-Ag sample with $SnO_2$ as the dielectric material and 10 nm of Ag).

**Table 1.** Deposition parameters for each material and the ion gun.

| Material | Target Purity (%) | % Weight | Power (W) | Power Density (W/cm$^2$) | Ar Flow (sccm) | O$_2$ Flow (sccm) | N$_2$ Flow (sccm) |
|---|---|---|---|---|---|---|---|
| $SnO_2$ | 99.99 | 100 | 2000 | 3.33 | 150 | 180 | 0 |
| ZnO | 99.99 | 100 | 1500 | 2.50 | 250 | 120 | 0 |
| AZO | 99.95 | Zn 98-Al 2 | 2000 | 3.33 | 300 | 0 | 0 |
| AZO_2 | 99.95 | Zn 98-Al 2 | 2000 | 3.33 | 270 | 30 | 0 |
| $SiAlN_x$ | 99.99 | Si 90-Al 10 | 2500 | 3.33 | 100 | 0 | 100 |
| Ag | 99.99 | 100 | 500 | 0.83 | 300 | 0 | 0 |
| Ti | 99.94 | 100 | 400 | 0.66 | 200 | 0 | 0 |
| Ion gun | --- | --- | 2 KV | --- | 50 | 0 | 0 |

**Table 2.** Composition and thickness of each layer deposited on samples S1–S12. Negative values indicate the thickness of the ion etching layer.

| Sample Type | Structure | Thickness (nm) | | |
|---|---|---|---|---|
| | | A (10 nm Ag) | B (15 nm Ag) | C (21 nm Ag) |
| S1 | Glass/$SnO_2$/Ag/Ti/$SnO_2$ | 29/10/2/45 | 24/15/2/49 | 33/21/2/55 |
| S2 | Glass/ZnO/Ag/Ti/ZnO | 29/10/2/45 | 24/15/2/49 | 33/21/2/55 |
| S3 | Glass/AZO/Ag/Ti/AZO | 29/10/2/45 | 24/15/2/49 | 33/21/2/55 |
| S4 | Glass/AZO_2/Ag/Ti/AZO_2 | 29/10/2/45 | 24/15/2/49 | 33/21/2/55 |
| S5 | Glass/$SiAlN_x$/Ag/Ti/$SiAlN_x$ | 29/10/2/45 | 24/15/2/49 | 33/21/2/55 |
| S6 | Glass/AZO/Ag/Ti/$SiAlN_x$ | 29/10/2/45 | 24/15/2/49 | 33/21/2/55 |
| S7 | Glass/$SiAlN_x$/AZO/Ag/Ti/$SiAlN_x$ | 14/14/10/2/45 | 24/15/2/49 | 16/16/21/2/55 |
| S8 | Glass/AZO/$SiAlN_x$/Ag/Ti/$SiAlN_x$ | 14/14/10/2/45 | 24/15/2/49 | 16/16/21/2/55 |
| S9 | Glass/$SiAlN_x$/AZO/Ag/Ti/$SiAlN_x$ | 24/5/10/2/45 | | 25/8/21/2/55 |
| S10 | Glass/AZO/Ion/Ag/Ti/$SiAlN_x$ | 39/-10/10/2/45 | | 43/-10/21/2/55 |
| S11 | Glass/$SiAlN_x$/Ion/Ag/Ti/$SiAlN_x$ | 39/-10/10/2/45 | | 43/-10/21/2/55 |
| S12 | Glass/$SiAlN_x$/AZO/Ion/Ag/Ti/$SiAlN_x$ | 14/19/-5/10/2/45 | | 16/21/-5/21/2/55 |

**Table 3.** Composition and thickness of each layer deposited on samples D1–D5.

| Sample Type | Structure | Thickness (nm) | |
|---|---|---|---|
| | | A (10 + 14 nm Ag) | B (7 + 19 nm Ag) |
| D1 | Glass/$SnO_2$/Ag/Ti/SnO2/Ag/Ti/$SnO_2$ | 29/10/2/77/14/2/34 | 30/7/2/78/19/2/32 |
| D2 | Glass/ZnO/Ag/Ti/ZnO/Ag/Ti/ZnO | 29/10/2/77/14/2/34 | 30/7/2/78/19/2/32 |
| D3 | Glass/AZO/Ag/Ti/AZO/Ag/Ti/AZO | 29/10/2/77/14/2/34 | 30/7/2/78/19/2/32 |
| D4 | Glass/AZO_2/Ag/Ti/AZO_2/Ag/Ti/AZO_2 | 29/10/2/77/14/2/34 | 30/7/2/78/19/2/32 |
| D5 | Glass/$SiAlN_x$/Ag/Ti/$SiAlN_x$/Ag/Ti/$SiAlN_x$ | 29/10/2/77/14/2/34 | 30/7/2/78/19/2/32 |



The first five samples for both single- and double-Ag coatings had different dielectric layers, that is, S1 and D1 contained $SnO_2$ and S2, D2 consisted of ZnO and S3, D3 contained AZO and S4, D4 contained AZO deposited with a small amount of $O_2$, and S5 and D5 contained $SiAlN_x$. These multilayers formed the basis of the study, as they can be used to compare the optical and energetic properties measured for each dielectric material. However, we also synthesized and assessed the performance of more complex structures. Sample S6 contained a combination of two dielectric materials, AZO in the first dielectric layer and $SiAlN_x$ in the final dielectric layer, as $SiAlN_x$ provides greater mechanical and thermal protection. Samples S7, S8, and S9 contained a combination of AZO and $SiAlN_x$ in different proportions and different arrangements in the first dielectric layer, and $SiAlN_x$ in the final dielectric layer, again, for mechanical and thermal protection. We decided to divide the first dielectric layer between AZO and $SiAlN_x$ in the hope that it would provide greater protection to the Ag layer during the thermal treatment, thus achieving greater thermal stability than with a single layer of AZO. An ion gun was used to etch the first dielectric layer in samples S10, S11, and S12. In S10 and S11, ion etching was applied to the first dielectric layer made of AZO or $SiAlN_x$, respectively, and in sample S12, the ion gun was used on the AZO layer that formed part of the dielectric combination in the first dielectric layer.

Hence, the structures were prepared with different dielectric materials and different amounts of silver so we could compare the emissivity and visible transmission of coatings with the same amount of silver.

### 2.2. Characterization

Layer thicknesses were measured individually with a DektakXT® mechanical profilometer, which has a precision of approximately 1 nm (see the profile of a single layer of $SnO_2$ in Figure S5 of the Supplementary Material). This measurement allows for the calculation of the exposure time required for each thickness of each material.

Optical measurements were made with a UV–Vis/NIR spectrophotometer designed and built by the Photonic Technologies Group at the University of Zaragoza. The spectrophotometer can perform specular transmission and reflection measurements across the entire range of the solar spectrum, from 300 to 2500 nm, with an angle of incidence of 8°. We used the following equations (1) and (2) to calculate the photoenergy factors that characterize low-emissivity and solar control coatings [62]:

$$T_{VIS} = \frac{\int_{380nm}^{780nm} T(\lambda)V(\lambda)D_{65}(\lambda)d\lambda}{\int_{380nm}^{780nm} V(\lambda)D_{65}(\lambda)d\lambda} \tag{1}$$

$$T_{SOLAR} = \frac{\int_{300nm}^{2500nm} T(\lambda)S(\lambda)d\lambda}{\int_{300nm}^{2500nm} S(\lambda)d\lambda} \tag{2}$$

where $T(\lambda)$ is the spectral transmittance factor (measured with the spectrophotometer), $V(\lambda)$ is the normalized spectral sensitivity curve of the human eye, $D_{65}(\lambda)$ is the standard illuminator, and $S(\lambda)$ is the solar spectrum. Analogous values can be calculated for the reflectance in a similar fashion.

Conductivity was determined by measuring the sample's sheet resistance using an SRM-12 surface resistance meter (NAGY Messsysteme GmbH, Gäufelden, Germany), which follows the eddy current technique (see Figure S6 of the Supplementary Material). Sample emissivity was calculated from the following Equation (3) [63]:

$$\varepsilon_n = 0.0106 \cdot R_\square \tag{3}$$

where $\varepsilon_n$ is the emissivity normal to the surface of the sample and $R_\square$ the sheet resistance. Equation (3) is a valid approximation for calculating emissivity in electrically conductive coatings, provided that $R_\square$ takes a low value, as observed in the structures studied here.

## 3. Results and Discussion

### 3.1. Emissivity

Figures 1–3 show the emissivities obtained for the S samples (single-Ag coatings) (see Table S2a–c of the Supplementary Material). The subtype A samples had emissivities of 0.083 with $SnO_2$ (S1A), 0.064 with ZnO (S2A), 0.058 with AZO (S3A), 0.063 with AZO deposited with a small amount of $O_2$ (S4A), and 0.067 with $SiAlN_x$ (S5A). The emissivities of the subtype B samples were 0.040 with $SnO_2$ (S1B), 0.031 with ZnO (S2B), 0.030 with AZO (S3B), 0.031 with AZO deposited with $O_2$ (S4B), and 0.035 with $SiAlN_x$ (S5B), while of the subtype C samples were 0.025 with $SnO_2$ (S1C), 0.020 with ZnO (S2C), 0.019 with AZO (S3C), 0.020 with AZO deposited with $O_2$ (S4C), and 0.023 with $SiAlN_x$ (S5C). As observed, the three subtypes of sample, A, B, and C, exhibited identical behavior with respect to the emissivity: The samples with the worst emissivities were those with $SnO_2$ dielectric layers, followed by $SiAlN_x$, while the ones with AZO had the best emissivities. Samples with ZnO and AZO deposited with $O_2$ achieved similar emissivities, even though they were formed from different targets. This is because both compounds incorporated Zn as the base material and were synthesized using oxygen in the deposition process, so they compacted in a similar manner during formation, resulting in very similar layers for silver deposition. Our results agree with those presented in [64], where the same sheet resistance of 7 Ω/sq (emissivity of 0.074) was reported for coatings of 9 nm of Ag with ZnO and 10 nm of Ag with $SnO_2$, which suggests that ZnO performs better than $SnO_2$. We can also compare our results with those published in the literature [4,33,36,54,65], where emissivity has been reported for tri-layer structures with dielectric layers of $SnO_2$ (0.12 for 10 nm of Ag), ZnO (0.024 for 17.7 nm of Ag), $SiN_x$ (0.03 for 15 nm of Ag), and AZO (0.064 for 10 nm of Ag). These results are similar to the ones obtained in our study. Therefore, AZO is the best dielectric material for applications that require a low emissivity, as silver growth is more efficient on AZO. Another effect that should also be considered is the influence of the roughness of the layers on emissivity. The superior smoothness of the layers enhances the emissivity of the entire structure, which is an effect that could be addressed in future fundamental studies, as our current study focuses on applied-level investigations.

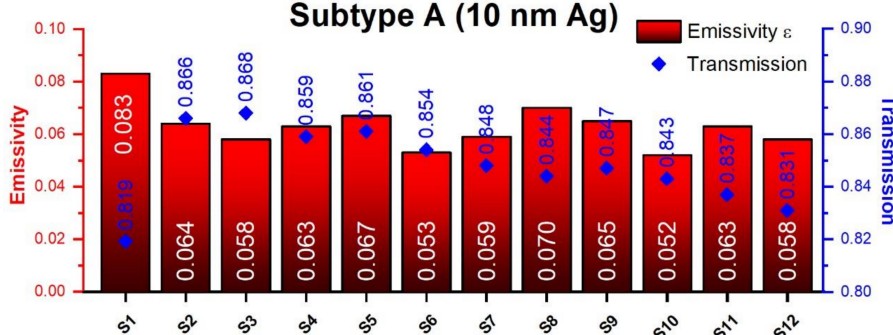

**Figure 1.** Emissivity and global visible transmission for subtype A samples with 10 nm Ag.

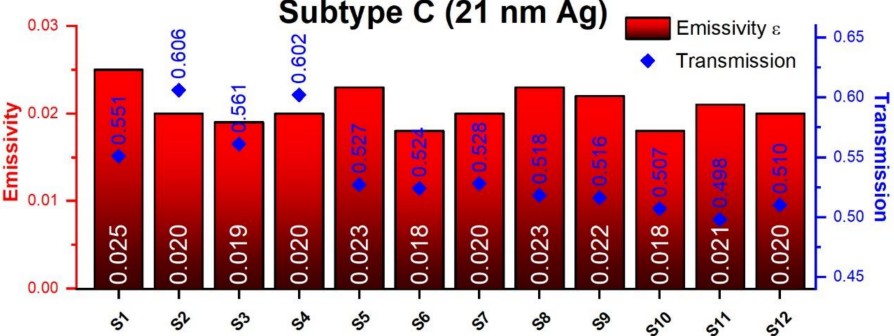

**Figure 2.** Emissivity and global visible transmission for subtype C samples with 21 nm Ag.

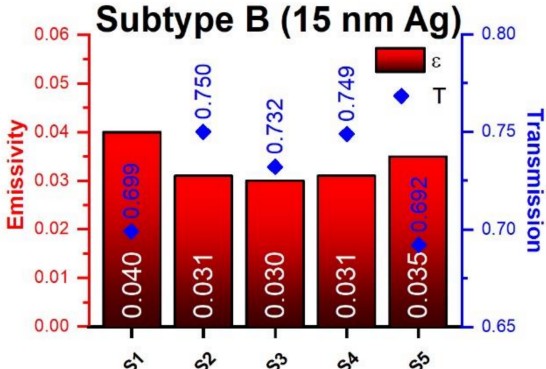

**Figure 3.** Emissivity and global visible transmission for subtype B samples with 15 nm Ag.

For subtype A and C samples, we also studied some more complex structures in addition to the aforementioned multilayers. They contained a combination of AZO (which, as we have seen, provides the best emissivity) and SiAlN$_x$ (for mechanical and thermal protection). Again, these samples presented the same behavior. The samples whose first dielectric layer was AZO and the second was SiAlN$_x$ (S6) exhibited a better emissivity than those with two AZO layers. According to [66], metal barrier layers, such as Ni, NiCr, or Ti, result in coatings with poorer emissivities. However, if the barrier layer is TiN$_x$, these materials exhibit better emissivity under certain deposition conditions, as stated in [67]. Therefore, when depositing SiAlN$_x$ after the thin Ti layer, a portion of the material transforms into TiN$_x$ due to the presence of N$_2$ in the chamber, and this TiN$_x$ contributes to better emissivity. Consequently, structures with SiAlN$_x$ as the final dielectric layer and a Ti barrier yield better emissivity compared to those with AZO as the last dielectric layer and a Ti barrier. This result is highly relevant to our research interests. Using SiAlN$_x$ as the final dielectric layer not only enhances the electrical properties of the structure, as we have just observed, but also contributes to improving the thermal stability of the entire structure [19]. Low-emissivity coatings with AZO dielectric exhibit superior thermal stability compared to those with SnO$_2$ and even surpass those with ZnO [55]. Therefore, sample S6, with AZO as the first dielectric layer and SiAlNx as the final one, holds promise as a noteworthy coating to consider.

The samples with the first dielectric layer divided between AZO and SiAlN$_x$ did not improve the emissivity obtained by sample S6. Moreover, better emissivity was achieved when silver was deposited onto an AZO dielectric layer rather than SiAlN$_x$ (S7 better than S8, as observed when the same dielectric material was used; S3 than S5). Furthermore, the emissivity was lower for greater thicknesses of AZO (S7 better than S9), as shown in [41,42]. As suggested in [42], this is because the crystallization of AZO is more uniform when depositing thicker layers. In turn, this improves the crystallization of silver and facilitates better spreading of silver over AZO, resulting in superior electrical properties for the entire structure. Finally, it is worth noting that samples that incorporated an ion etching step, both on AZO and SiAlN$_x$, had lower emissivities than those that did not (S10 outperformed S6, S11 was lower than S5, and S12 was lower than S9). This agrees with the findings reported in [60], which studied the effect of ion gun etching on the SnO$_2$ dielectric layer of low-emissivity structures. The authors concluded the lower emissivities were due to the reduced roughness of the SnO$_2$ layer before Ag deposition, in accordance with our study.

Figures 4 and 5 show the emissivities obtained for the D samples (double-Ag coatings) (see Table S3a,b of the Supplementary Material). Subtype A samples had emissivities of 0.031 with SnO$_2$ (D1A), 0.023 for ZnO (D2A), 0.024 with AZO (D3A), 0.023 for AZO deposited with O$_2$ (D4A), and 0.025 with SiAlN$_x$ (D5A). The subtype B samples yielded values of 0.028 for SnO$_2$ (D1B), 0.020 with ZnO (D2B), 0.021 with AZO (D3B), 0.020 for AZO deposited with O$_2$ (D4B), and 0.023 with SiAlN$_x$ (D5B). As with the single-Ag samples, the structures with AZO, ZnO, and AZO deposited with O$_2$ exhibited the best performance and with similar emissivities. The structures had exceptionally low emissivities, which

made it hard to detect any differences between them as they were in the same magnitude as the measurement accuracy. However, the difference was more discernible when the silver content was reduced, as observed in the subtype SA structures (10 nm Ag). Once again, $SnO_2$ (sample D1) was the dielectric material with the highest emissivity among those studied here.

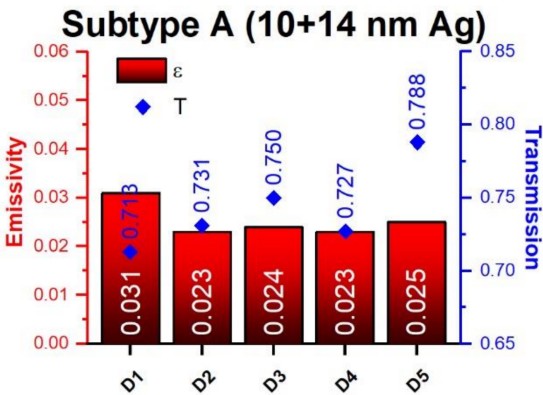

**Figure 4.** Emissivity and global visible transmission for subtype A samples with 10 + 14 nm Ag.

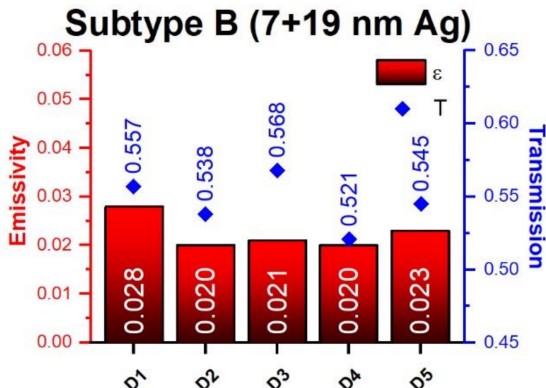

**Figure 5.** Emissivity and global visible transmission for subtype B samples with 7 + 19 nm Ag.

### 3.2. Optical Properties

All structures under study were designed with approximately the same thickness of dielectric material. For the same multilayer structure, the maximum global visible transmission occurs when the maximum spectral transmittance is at approximately 550 nm, the wavelength to which the human eye is most sensitive. All materials used in low-e coatings have a refractive index between 1.85 and 2.10 at 550 nm, with the following values reported in different studies: 2.019 for ZnO [68], 2.023 for $SiN_x$ [69], 1.85 for AZO [70], 2.00 for $SnO_2$ [71], and 2.09 for $SiAlN_x$. Dielectric materials with different refractive indexes create different optical paths, and this interference can shift the maximum spectral transmission. Therefore, even if the displacement is small, it must be considered when assessing the global visible transmission, because if the maximum spectral transmission is displaced, the global visible transmission could easily be improved by adjusting the thicknesses of the dielectric layers. It is also worth noting that the deposition rate can be subtly affected by target wear or changes in the base pressure of the process chamber.

Another factor that can have a strong influence on the visible optical properties of the structures presented in this study is the Ti barrier layer, which is an absorbent metal. This thin layer protects the Ag and prevents its oxidation when depositing the dielectric layer by means of reactive sputtering with $O_2$. In this scenario, the Ti barrier layer becomes slightly oxidized and its optical absorption decreases. Thus, the barrier layer would have to be adjusted to each specific process to optimize the oxidation level of the Ti layer, i.e.,

it must maintain its barrier function while also reducing its optical absorption through a suitable oxidation level.

It is also important to consider the influence of the roughness of the layers on transmittance. The smoother the surface, the less light will scatter and the higher its transmittance. This is an effect that, as we discussed in the emissivity section, could be taken into account for future studies.

The type of substrate used for layer deposition also contributes to improving or worsening the visible transmission of the structure. In this study, low-iron glass was utilized for its high transparency, low absorption, and good smoothness, which aids in layer adhesion. Other substrates, such as plastics or lower-performing glasses, could be used, but they would lead to a corresponding deterioration in the optical properties of the layers under investigation. In agriculture, low-emissivity coatings have been employed on plastics for use in greenhouses, but that is another application outside the scope of our study.

Figures 1–3 show the global visible transmission for single-Ag samples calculated according to DIN EN410 [62], while Figure 6 shows the visible spectrum in transmittance for subtype A, B, and C samples. The low-emissivity coatings with $SnO_2$ (S1) had a much lower transmittance than the structures prepared with other dielectric materials. This is most evident in Figure 6a,c, where the maximum values for subtypes A and C associated with $SnO_2$ were comparatively lower. The ZnO and AZO coatings (S2, S3, and S4) generally had the highest transmittance with a similar performance, because they are similar materials after all. This is also evidence of the excellent repeatability in the manufacture of samples in this study. In the case of $SiAlN_x$ (S5), distinct transmittance values have been obtained. While the transmittance for subtype A was similar for ZnO and AZO structures, it was noticeably lower for subtypes B and C, even slightly inferior to the samples with a $SnO_2$ dielectric layer. A notable characteristic of the $SiAlN_x$ structures was their elevated transmittance in the near UV range, specifically between 300 and 400 nm. This could be quite an important feature, as it could be an advantage or disadvantage in low-emissivity coatings, depending on the specific requirements.

Samples with more complex structures in subtypes A and C (S6 to S12, which had $SiAlN_x$ and AZO in their structure) all had quite similar visible transmissions, but none of them were an improvement on the structures made with ZnO or AZO (S2, S3, and S4). Nevertheless, the S6 sample exhibited a visible transmission comparable to that of S5, but with better emissivity. This is a very interesting quality because these samples, as they have higher thermal resistance, could withstand subsequent manufacturing processes, such as tempering or bending.

Figures 4 and 5 show the global visible transmission for double-Ag samples calculated according to DIN EN410 [62], while Figure 7 shows the visible spectrum in transmittance for subtype A and B samples. Here it is notable that the sample with $SnO_2$ in its structure (D1) had the lowest transmittance among subtype A. However, for subtype B, the transmittance was comparable to that of the other materials. This is because the maximum spectral transmittance of this particular structure was at 550 nm, as shown in Figure 7, while the other structures exhibited maximum spectral transmittances of over 550 nm, albeit at a different wavelength (approx. 500 nm). Again, the ZnO and AZO samples (D2, D3, and D4) had better visible transmission, so they are suitable for use in low-e coatings. Finally, it is worth highlighting the results of the $SiAlN_x$ samples (D5), as they were similar to the results of ZnO and AZO in subtype B and much better in subtype A. Figure 7a reveals that sample D5A had a higher spectral transmittance across the entire visible range than the rest of the samples. This was a surprising result and indicates that while using oxide-based dielectric materials we achieved consistent and repetitive results; in the case of $SiAlN_x$, we obtained better results in some cases and worse in others. This translates into lower process repeatability, so we believe this material should be studied further in an attempt to optimize the deposition conditions and determine how to repeat the good results obtained with sample D5A. We suspect that crucial factors in this type of structure are the oxidation

and diffusion processes occurring within the Ti barrier layer, as less oxidation in this layer may lead to significant optical absorption.

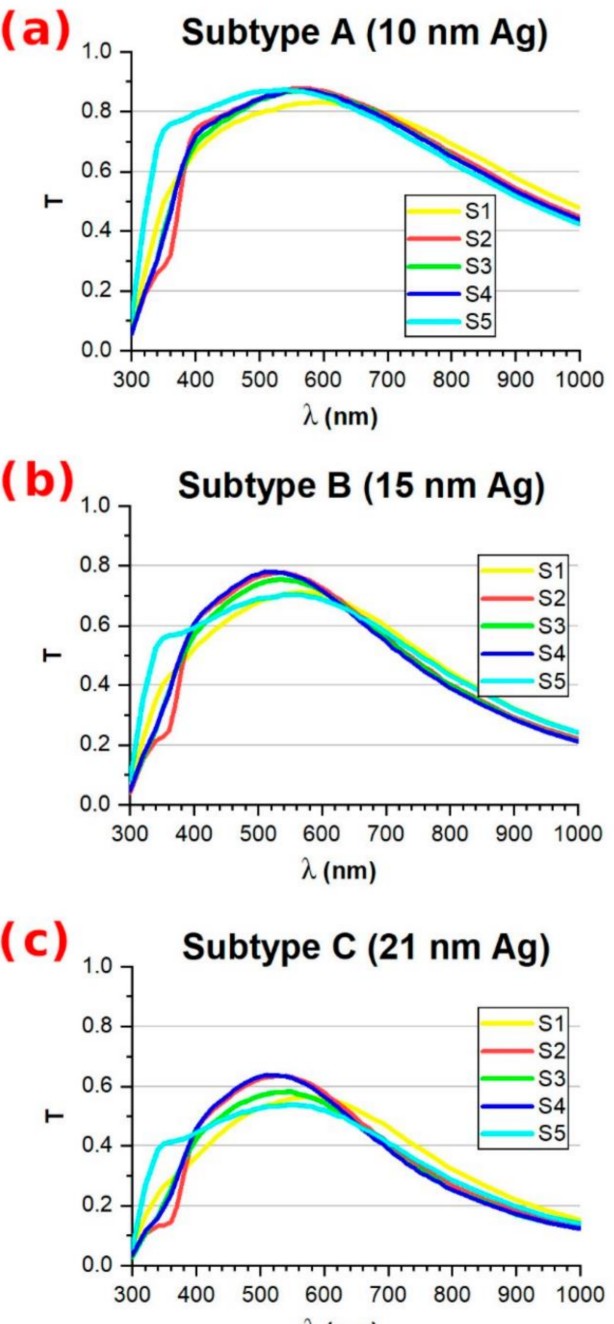

**Figure 6.** Spectral visible transmission for (**a**) subtype A samples with 10 nm Ag, (**b**) subtype B samples with 15 nm Ag, and (**c**) subtype C samples with 21 nm Ag.

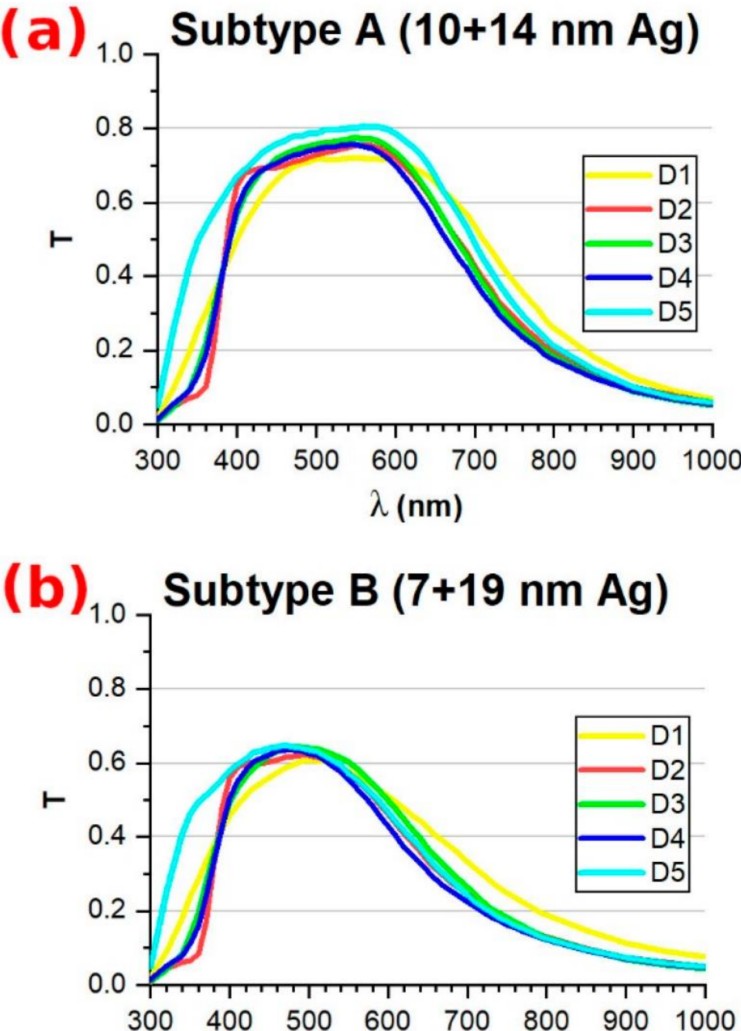

**Figure 7.** Spectral visible transmission for (**a**) subtype A samples with 10 + 14 nm Ag and (**b**) subtype B samples with 7 + 19 nm Ag.

## 4. Conclusions

This work carried out a systematic study of the optical and energetic properties of low-emissivity structures incorporating silver and various dielectric materials. Based on the results, we can conclude that AZO demonstrated the lowest emissivities, showing, in the case of 10 nm Ag coatings, an improvement in the emissivity from 0.083 with $SnO_2$, 0.064 with ZnO, 0.063 with AZO deposited with $O_2$, and 0.067 with $SiAlN_x$, to 0.058 with AZO. This is attributed to the superior efficiency of silver growth on AZO. Furthermore, when the multilayer comprised the same structure as the silver layer, we obtained lower emissivities for increasing thicknesses of the AZO layer onto which the silver was deposited, worsening emissivity values from 0.053 to 0.059 when the AZO layer was reduced by half and to 0.065 when it was reduced by three-quarters for 10 nm Ag coatings. This is because a thicker AZO layer exhibits improved crystallization during the deposition process, facilitating enhanced crystal formation and the distribution of silver on its surface. Consequently, this translates into improved electrical properties throughout the entire structure. These results were confirmed for different silver thicknesses and for both single-Ag structures and double-Ag structures. Similarly, etching the dielectric surface with an ion gun before Ag deposition improved the emissivity of the resulting structures, as ion etching smooths the surface and eliminates any roughness, thus improving the subsequent processes of

silver deposition and growth. We achieved improvements in the emissivity when the ion gun was used over AZO from 0.053 to 0.052 and over SiAlNx from 0.067 to 0.063, compared to depositing the dielectric without using an ion gun, for 10 nm Ag coatings.

We also examined the global visible transmission for different low-e structures with the same dielectric materials. Of the materials studied here, the oxide-based dielectric layers offered the highest visible transmission in our deposition system, showing, in the case of coatings with ZnO and 21 nm of Ag, an improvement in the visible transmission from 56.1% with AZO, 60.2% with AZO deposited with $O_2$, and 52.7% with $SiAlN_x$, to 60.6% with ZnO. This was especially true for single-Ag structures, where the deposition process caused slight oxidation of the Ti barrier layer, resulting in reduced absorption. In the case of the double-Ag structures, however, the nitrogen-based $SiAlN_x$ dielectric layer achieved the highest transmission. Future studies could look into this intriguing phenomenon to determine the causes of this behavior.

(See Conclusion Highlights included in the Supplementary Material).

**Supplementary Materials:** The following supporting information can be downloaded at https://www.mdpi.com/article/10.3390/coatings13101709/s1. Table S1: Abbreviations that appear in the manuscript with their meaning; Figure S1a: Photograph of the process chamber showing the ion gun and magnetrons; Figure S1b: Scheme of the deposition system; Figure S2: Sn target; Figure S3: Photographs of the labels of two of the targets; Figure S4: Cross-section of Sample S1B (coating completed) performed by FE-SEM; Figure S5: Profile of a single 92 nm $SnO_2$ layer; Figure S6: Photograph of the NAGY SRM-12 instrument for the measurement of sheet resistance; Table S2a: Properties of simple Ag samples with 10 nm Ag; Table S2b: Properties of simple Ag samples with 15 nm Ag; Table S2c: Properties of simple Ag samples with 21 nm Ag; Table S3a: Properties of double-Ag samples with 24 nm Ag; Table S3b: Properties of double-Ag samples with 26 nm Ag; Conclusion Highlights.

**Author Contributions:** Conceptualization, methodology, investigation, and writing—review and editing, A.C. and E.C.; data curation and writing—original draft preparation, A.C.; supervision, project administration, and funding acquisition, E.C. All authors have read and agreed to the published version of the manuscript.

**Funding:** This research was funded by the Spanish Ministerio de Ciencia e Innovación, under the project RTC2019-007368-3 and the Departamento de Ciencia, Universidad y Sociedad del Conocimiento del Gobierno de Aragón, under the group T20_20R.

**Institutional Review Board Statement:** Not applicable.

**Informed Consent Statement:** Not applicable.

**Data Availability Statement:** Data will be made available upon request.

**Acknowledgments:** The authors gratefully acknowledge the continued support from the company Aniño Duglass and Cátedra Ariño Duglass.

**Conflicts of Interest:** The authors declare no conflict of interest.

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
