# Peer review of "Comparison of the Optical Properties of Different Dielectric Materials (SnO2, ZnO, AZO, or SiAlNx) Used in Silver-Based Low-Emissivity Coatings"

_coatings, doi:10.3390/coatings13101709_

Round 1
Reviewer 1 Report
In this study, the authors compared the the optical properties of different dielectric materials (SnO2, ZnO, AZO or SiAlNx) used in silver-based low emissivity coatings. The manuscript will be accepted after addressing the following issues;
1) There are too much abbreviations are used in this article. I recommend to make a table to introduce all the abbreviations so that reader understand easily.
2) What you did in this article should be written in the abstract. Reader is confuse what you actually did or what is your part in this part?
3) There is no information related to source of the material and their purifications etc.
4) The authors mentioned the thickness of the different layers but there is no evidence. The authors should provide SEM or TEM image to confirm the thickness
5) The authors grown the thin film of different dielectric materials but the authors did not provide any morphology structure etc.
6) What’s about the roughness of the thin films because the absorbance or transmittance of the light based on the roughness?
7) What is the effect of the substrate on the optical properties, if it is replaced with other materials?
8) Many spelling and formatting typos in this paper, and the authors should check and revise them thoroughly
Many spelling and formatting typos in this paper, and the authors should check and revise them thoroughly
Reviewer 2 Report
The paper presented is interesting but the Authors should pay attention to several issues:
- the abstract should be supplemented with a brief summary of the results of the analysis
- the introduction should be edited emphasizing more the novelty of the work
- there seems to be an editorial error in equation 3
- Figures 1-3 should have more standardized ranges on the axes to make it easier to compare values
- the most important conclusions should be bulleted to make it easier for the reader to analyze them
Reviewer 3 Report
This manuscript reports on a comparison of the optical properties of different dielectric materials (SnO2, ZnO, AZO or SiAlNx) employed in silver-based low emissivity coatings. The purpose of the study is to investigate and compare the optical and photoenergetic properties of low emissivity coatings made from various dielectric materials deposited through magnetron sputtering. The authors claim that aluminum-doped zinc oxide (AZO, in fact a semiconductor) unifies best the optical and photoenergetic properties as per the chosen selection and deposition technique/system.
The manuscript provides a clear motivation and all the necessary experimental details, while taking a very original point of departure to the mentioned material systems. This work is also very timely. As a result, in my opinion, the present manuscript has good potential to inspire further studies and to attract a significant number of future citations.
Characterization techniques (including the important for this study UV–Vis/NIR) are well chosen and the characterization results are carefully presented and discussed thus convincingly substantiating the authors’ claims.
Discussion is easy enough to follow, well-structured, and accompanied by comprehensive figures of mostly good quality.
I recommend that this manuscript is accepted in Coatings after acknowledging the following points (amounting to “minor revision”):
1. Title could possibly be shortened avoiding the wording “different” and “used”.
2. Abstract: Characterization techniques should be clearly stated in the abstract.
3. The Introduction is well-written. However, understanding the physics behind complex materials obtained by magnetron sputtering can be efficiently assisted by theoretical methods (e.g., Journal of Physics: Condensed Matter 29 (2017) 195701) while optical properties are best addressed by time-dependent DFT calculations (e.g., The Journal of Physical Chemistry C 118 (2014) 11377-11384. These aspects should be acknowledged.
4. The thermal stability of the coatings (especially AZO) of interest should be mentioned/discussed in more detail.
5. Conclusions should include 2-3 of the most important quantitative results obtained from characterization.
Stylistic and grammatical revision of the text is needed. Some too long sentences are noticeable throughout the text.
Round 2
Reviewer 1 Report
accepted in the present form
Minor editing of the English language required